# Evaluating Back-to-Back and Day-to-Day Reproducibility of Cortical GABA+ Measurements Using Proton Magnetic Resonance Spectroscopy (^1^H MRS)

**DOI:** 10.3390/ijms24097713

**Published:** 2023-04-23

**Authors:** Sonja Elsaid, Peter Truong, Napapon Sailasuta, Bernard Le Foll

**Affiliations:** 1Translation Addiction Research Laboratory (TARL), Centre for Addiction and Mental Health (CAMH), Toronto, ON M5S 2S1, Canada; 2Institute of Medical Science (IMS), Faculty of Medicine, University of Toronto, Toronto, ON M5S 1A8, Canada; 3Brain Health Imaging Centre (BHIC), Centre for Addiction and Mental Health (CAMH), Toronto, ON M5T 1R8, Canada; 4Sunnybrook Health Science Centre, Sunnybrook Research Institute, Toronto, ON M4N 3M5, Canada; 5Departments of Tropical Medicine, Medical Microbiology and Pharmacology, John A. Burns School of Medicine, University of Hawaii, Honolulu, HI 96813, USA; 6Department of Psychiatry, University of Toronto, Toronto, ON M5T 1R8, Canada; 7Department of Pharmacology and Toxicology, Faculty of Medicine, University of Toronto, Toronto, ON M5S 1A8, Canada; 8Department of Family and Community Medicine, University of Toronto, Toronto, ON M5G 1V7, Canada; 9Addictions Division, Centre for Addiction and Mental Health (CAMH), Toronto, ON M6J 1H4, Canada; 10Waypoint Centre for Mental Health Care, Waypoint Research Institute, Penetanguishene, ON L9M 1G3, Canada

**Keywords:** γ-aminobutyric acid (GABA), proton magnetic resonance spectroscopy (^1^H MRS), reproducibility, healthy volunteers, dorsomedial pre-frontal cortex (dmPFC), anterior cingulate cortex (ACC), dorsolateral pre-frontal cortex (dlPFC), sex-differences

## Abstract

γ-aminobutyric acid (GABA) is a major inhibitory neurotransmitter implicated in neuropsychiatric disorders. The best method for quantifying GABA is proton magnetic resonance spectroscopy (^1^H MRS). Considering that accurate measurements of GABA are affected by slight methodological alterations, demonstrating GABA reproducibility in healthy volunteers is essential before implementing the changes in vivo. Thus, our study aimed to evaluate the back-to-back (B2B) and day-to-day (D2D) reproducibility of GABA+ macromolecules (GABA+) using a 3 Tesla MRI scanner, the new 32-channel head coil (CHC), and Mescher–Garwood Point Resolved Spectroscopy (MEGA-PRESS) technique with the scan time (approximately 10 min), adequate for psychiatric patients. The dorsomedial pre-frontal cortex/anterior cingulate cortex (dmPFC/ACC) was scanned in 29 and the dorsolateral pre-frontal cortex (dlPFC) in 28 healthy volunteers on two separate days. Gannet 3.1 was used to quantify GABA+. The reproducibility was evaluated by Pearson’s r correlation, the interclass-correlation coefficient (ICC), and the coefficient of variation (CV%) (r/ICC/CV%). For Day 1, B2B reproducibility was 0.59/0.60/5.02% in the dmPFC/ACC and 0.74/0.73/5.15% for dlPFC. For Day 2, it was 0.60/0.59/6.26% for the dmPFC/ACC and 0.54/0.54/6.89 for dlPFC. D2D reproducibility of averaged GABA+ was 0.62/0.61/4.95% for the dmPFC/ACC and 0.58/0.58/5.85% for dlPFC. Our study found excellent GABA+ repeatability and reliability in the dmPFC/ACC and dlPFC.

## 1. Introduction

γ-aminobutyric acid (GABA) is a major inhibitory neurotransmitter in the mammalian brain. Together with glutamate and glutathione, GABA is linked to metabolic pathways in the brain [1,2]. GABA is synthesized through the decarboxylation of glutamate by two isozymes of glutamic acid decarboxylase (GAD) 65 and GAD 67 [3]. It acts through ionotropic GABA_A_ and metabotropic GABA_B_ receptors, involved in the neuromodulation of inhibitory and excitatory brain currents [4,5]. Aside from binding to GABA receptors at the synapse, GABA is transported to glial cells. Inside glial cells, it enters GABA shunt via GABA transaminase (GAT) directly into the Krebs cycle, the key metabolic pathway for cellular energy production [1,6]. From the Krebs cycle, glutamate is synthesized and converted into GABA, completing the energy supply cycle in the GABA shunt. Additionally, glutamate is one of the three amino acids needed to synthesize glutathione, which is essential for the degradation and synthesis of proteins and protects the cell against reactive oxygen species [1,6,7]. Due to the metabolic linkage, the concentrations between GABA, glutamate, and glutathione are related [1,6].

GABA concentrations in the brain may vary depending on physiological states [8]. For instance, GABA changes with the menstrual cycle and is reduced during the luteal phase relative to the follicular phase [8,9]. It also varies with age [10], while some evidence shows that GABA may vary between sexes [11]. Some foods, such as whole grains, soy, fish, tomatoes, spinach, and broccoli, contain higher amounts of GABA or may boost its production in the body [12]. Caffeine decreases GABA in the brain, whereas alcohol and nicotine increase it [13,14,15]. Thus, the effects of diet and physiological changes on GABA must be controlled in vivo.

Dysregulation of GABA has been implicated in acute altered psychological states [16,17], the pathogenesis of many neuropsychiatric disorders, including epilepsy [18,19], schizophrenia [20,21], anxiety disorders [22,23,24], and after traumatic brain injury [6]. Explicitly, the imbalances between inhibitory–excitatory neurocircuitry have been noted in the brain.

For example, lower GABA levels have been detected in individuals experiencing higher stress levels [16,17], social anxiety disorder [22,23], and epilepsy [18,19]. Alternatively, the administration of pharmacological treatments for these disorders [20,22,24] (that led to symptom relief) was associated with GABA normalization, indicating the importance of measuring GABA as a marker of treatment efficacy. Following traumatic brain injury, disturbances in energy metabolism led to decreases in glutamate and increases in GABA and glutathione levels [2,25,26]. Since glutathione is synthesized from glutamate, depletions in glutamate may be due to increased glutathione synthesis to mitigate oxidative stress. Increased GABA may be a compensatory mechanism activated to reduce neuronal hyperactivity [2,25,26].

Proton magnetic resonance spectroscopy (^1^H MRS) offers a non-invasive way to measure human brain metabolites in vivo. ^1^H MRS reports on magnetic resonance signals from protons in molecules [27]. GABA has six observable protons arranged in three methylene groups resonating at three different frequencies; two triplets (resonating at 3.01 ppm and 2.28 ppm) and a quintuplet centered at 1.89 ppm [27]. However, several challenges exist with detecting GABA. First, the concentration of GABA in the brain is ~1 mM, which is too low to accurately detect using MR scanners with lower magnetic fields (<7T) [27,28]. Second, GABA’s spectral peaks are overlapped by much larger resonances, such as total creatine (tCr); N-acetyl-L-aspartate (NAA); glutamate (Glu); and glutamine (Gln) [8,29]. One way to overcome these challenges is using J-difference editing approaches, such as the Mescher–Garwood Point Resolved Spectroscopy (MEGA-PRESS) technique [30]. However, it is essential to note that due to the proximity of macromolecule (MM) resonance (located at 1.7 ppm) to the frequency selective radiofrequency (RF) inversion (or editing) pulse at 1.9 ppm, the MM signal is also coedited alongside GABA [8,27,29]. Thus, the resulting signal is called GABA+, containing both GABA and MM. In addition, increasing the sensitivity of GABA+ detection can be achieved by increasing the number of phase array coils [31] and optimizing acquisition parameters to the values that would reduce the duration of single scan sessions while maximizing the quality of GABA+ measurement [32,33]. Considering that many physiological and technical factors (e.g., MRS sequence parameters) affect the accuracy of GABA+ measurements, conducting reproducibility studies on healthy volunteers has become an essential step in demonstrating the efficacy of specific GABA+ determining methods before using them in vivo. The best way to conduct these studies is to examine back-to-back (B2B) and day-to-day (D2D) test–retest reproducibility of GABA+. The purpose of B2B scan assessments is to evaluate the reproducibility of the technical variables (e.g., hardware, software-related, MRS sequence parameters) while attempting to keep the physiological variations minimal. The scan time and spectroscopic parameters determined in B2B studies could be used in single-scan clinical assessments or case studies. The aim of comparing the reproducibility parameters of scans conducted on different days is to capture the effects of participants’ daily physiological variations and technical stability. Since daily physiological variations are assumed in D2D reproducibility experiences, the scan time and spectroscopic parameters from these experiments could be used in clinical investigations necessitating repeated patient MRS assessments or cross-sectional or longitudinal clinical experiments.

Patients with psychiatric symptoms often experience discomfort if asked to undergo MRI scanning procedures for longer periods (more than 10 min) [34]; thus, the general aim of MRS reproducibility studies is to find spectroscopic parameters utilizing the shortest possible scan times. Previous research demonstrated adequate B2B reproducibility with a 10 min scan time, using the 3 Tesla (T) scanner and the 64-channel head coil (CHC) in the anterior cingulate cortex (ACC) and dorsolateral pre-frontal cortex (dlPFC) [35]. Moreover, D2D reproducibility with 13—and 8.4 min—scan times were shown with the 3T scanner and 32 CHC in the ACC and dlPFC, respectively [36,37].

With the growing demand for further reduction in the MRS scan times, our study aims to examine the B2B and D2D test–retest reproducibility of GABA+ using the 3T scanner, 32 CHC, and 5.2 min scan time. In this study, the dorsomedial pre-frontal cortex (dmPFC)/ACC and dlPFC regions of interest (ROI) will be scanned in healthy volunteers (age range 20–32), and spectroscopy data will be processed with a user-friendly software, Gannet 3.1 [38]. Our exploratory analysis assessed within- and between-sex differences in GABA+ concentrations, considering mixed evidence in GABA+ concentration between male and female participants [11].

## 2. Results

### 2.1. Within Session/B2B Reproducibility

#### 2.1.1. dmPFC/ACC

Based on the exclusion criteria, only one GABA+ data point was removed from the analysis after determining that GABA+ full width at half maximum (FWHM) for this point was 3 standard deviations (SD) above the mean. The reproducibility parameters for both Day 1 and Day 2 B2B scans are displayed in Table 1. Pearson’s correlations are displayed in Figure 1, whereas Bland–Altman plots are shown in Figure 2a,b. Within-subject CV% mean differences are plotted in Figure 3a.

According to the Bland–Altman plot analysis, only 2/28 data points for Day 1 B2B (points a and b in Figure 2a) and only 1/29 data points for the Day 2 B2B comparison (point c in Figure 2b) were located above the upper limit of agreement (bias + 1.96 SD).

#### 2.1.2. dlPFC

Four data points were removed from the Day 1 analysis as they did not meet the criteria for spectral quality. The first data point could not be fitted by Gannet software. GABA+ fit errors % were more than 3SD above the mean for the second and third data points, while a noisy edited GABA+ spectrum (scan 1 for Day 1) was observed for the fourth. Moreover, three data points were removed from the Day 2 B2B analysis. In this case, the first data point could not be fitted by Gannet. For the second point, GABA+ fit error % was more than 3SD above the mean, and for the third, pre-post processing misalignment of creatine frequencies was observed in the GannetLoad output. The reproducibility parameters for both Day 1 and Day 2 B2B scans are displayed in Table 2. Pearson’s correlations are displayed in Figure 1c,d, whereas Bland–Altman plots are shown in Figure 2c,d. Within-subject CV% differences are plotted in Figure 3b.

### 2.2. D2D Reproducibility of 5.2 Min Scans

#### dmPFC/ACC

Four comparisons were made when analyzing the D2D reproducibility of 5.2 min scans. The following comparisons were made: (1) S1D1 and S1D2; (2) S2D1 and S2D2; (3) S1D1 S2D2; and (4) S2D1 and S1D2. The reproducibility parameters for all four comparisons are listed in Table 3. Pearson’s correlation plots are shown in Appendix A, Bland–Altman plots are shown in Appendix A, whereas the within-subject CV% mean differences are displayed in Figure 3a.

### 2.3. dlPFC

Similarly to the dmPFC/ACC, four sets of 5.2 min scan D2D reproducibility comparisons were made as in the previous section. The reproducibility parameters for all four comparisons are listed in Table 4. Pearson’s correlation plots are shown in Appendix A, whereas Bland–Altman plots are indicated in Appendix A. Within-subject CV% means are displayed in Figure 3b.

### 2.4. D2D Reproducibility of Two Averaged B2B 5.2 Min Scans

In anticipation of lower-than-expected D2D reproducibility when comparing four 5.2 min scans, we averaged two Day 1 and Day 2 B2B scans to produce GABA+ concentrations equivalent to a 10.4 min scan. The reproducibility parameters for the comparison of Day 1 and Day 2 GABA+ averaged values are shown in Table 5. Pearson’s correlation plots for the dmPFC and dlPFC averaged D2D comparisons are plotted in Figure 4, while Bland–Altman plots are shown in Figure 5.

### 2.5. Assessment of D2D Voxel Overlaps, Tissue Heterogeneity, and GABA+ Concentrations

Considering that the voxels were repositioned onto T1-weighted images on the second scanning day, we computed % of voxel overlaps between Day 1 and Day 2 scanning sessions. Furthermore, we determined the correlation between voxel overlap percentages and % mean differences between Day 1 and Day 2 averaged GABA+ concentrations. This comparison aimed to assess whether disparities in voxel overlaps were related to D2D variations observed with our data. For the dmPFC/ACC, voxel overlap was 89.3% ± 4.0%, with a range of (79.0–95.1%), whereas for dlPFC, it was 75.7% ± 13% (40.9–97.6%). The correlation between voxel overlaps and GABA+ values described above was r = −0.37 and *p* = 0.051 for the dmPFC/ACC, and r = −0.07 and *p* = 0.731 for the dlPFC.

For each ROI, the tissue composition—a fraction of the gray matter/white matter/cerebrospinal fluid (GM/WM/CSF)—was determined and compared between Day 1 and Day 2 sessions to investigate its effect on GABA+ reproducibility parameters. Tissue heterogeneity was consistent between the sessions. Table 6 shows tissue heterogeneity for each voxel. GABA+ concentrations are listed in Appendix A for the dmPFC/ACC and in Appendix A for the dlPFC.

### 2.6. Sex Differences

The investigation of GABA+ concentration sex differences was conducted using Day 1 and Day 2 averaged data, provided that averaged, rather than individual, 5.2 min scan parameters produced better D2D reproducibility results.

For the dmPFC/ACC, 14 females (with Day 1 GABA+ concentrations at 1.91 ± 0.21 i.u. and for Day 2 1.92 ± 0.24) were compared to 14 males (with Day 1 GABA+ concentrations at 1.89 ± 0.14 i.u. and for Day 2, 1,90 ± 0.21). The comparison of mean GABA+ values between males and females on Day 1 and Day 2 showed no significant differences. For Day 1 analysis, t = 0.30 and *p* = 0.767, whereas for Day 2, t = 0.14 and *p* = 0.887. Similarly, the investigation of within-sex variation in GABA+ concentrations between Day 1 and Day 2 parameters demonstrated no significant findings (t = −0.03 and *p* = 0.980 for females and t = −0.24 and *p* = 0.409 for males).

For dlPFC, similar findings were also observed. The analysis showed that on Day 1, t = −0.96 and *p* = 0.348, and on Day 2, t = −0.12 and *p* = 0.907 when comparing GABA+ concentrations of 13 females (Day 1 GABA+ was 1.25 ± 0.14 i.u. and Day 2 was 1.26 ± 0.09 i.u.) to 10 males (Day 1 GABA+ was 1.19 ± 0.16 i.u. and Day 2 was 1.25 ± 0.21 i.u.). Within-sex analysis between Day 1 and Day 2 averaged scans revealed the following statistical parameters: t = −0.18 and *p* = 0.863 for females and t = −1.16 and *p* = 0.277 for males. Between- and within-sex differences are shown in Figure 6a for the dmPFC/ACC and Figure 6b for the dlPFC.

## 3. Discussion

This study aimed to determine the B2B and D2D test–retest reproducibility of GABA+ in the dmPFC/ACC and the dlPFC in healthy volunteers. Our exploratory analysis entailed using the averaged Day 1 and Day 2 data (equivalent to 10.4 min/scan) to assess between and within sex differences in GABA+ concentrations. Below, we discuss our findings for each type of comparison.

### 3.1. B2B Reproducibility

For both ROIs, B2B comparison of two single-session scans indicated good repeatability (CV% of 5.02–6.26% for the dmPFC/ACC and CV% of 5.15–6.89% for the dlPFC), a sizeable inter-scan correlation between GABA+ concentrations on both assessment days, and moderate to good reliability (ICC of 0.59–0.60 for the dmPFC/ACC and 0.54–0.73 for the dlPFC). Our B2B results are comparable to two other studies investigating the B2B GABA+ reproducibility in the ACC and dlPFC, in which slightly different study parameters were used [30,31].

In the study by Duda et al., 2020, a Siemens 3T Prisma scanner (Siemens Healthcare, Erlangen, Germany) with 64 CHC (manufacturer of the head coil not specified) and MEGA-PRESS with echo time (TE) = 68 ms; repetition time (TR) = 3000 ms with 192 acquisition averages were used. B2B scans of the generated within-subject repeatability of 7.5% for the ACC and 3.97% for dlPFC (CV%), moderate reliability (ICC = 0.44) in the ACC and excellent reliability in the dlPFC (0.87) [35]. Given that 64 was used rather than 32 CHC in this study, increased SNR gains (by the square root of 2) [31] were expected. Thus, observing slightly better reproducibility parameters in the dlPFC in this study was not surprising. Contrary to these findings, our study observed better reproducibility results with ACC ROI (despite more considerable SNR gains in the study by Duda et al. 2020), possibly because of two crucial study differences. First, we used a larger 24 mL ACC voxel, compared to 17.5 mL in the other study. Using larger voxel sizes allows for more signal\ to be acquired in a single average; thus, it will produce higher SNR, potentially leading to a more accurate determination of metabolite concentrations and better reproducibility parameters [32]. Second, Duda and colleagues used rostral rather than dmACC. Rostral ACC is anatomically located near the sinus cavity, filled with air. Therefore, this voxel proximity may have contributed to poor shim quality, ultimately affecting the accuracy of metabolite measurements in this brain region [35].

Mikkelsen and colleagues used a 3T GE Signa HDx MRI scanner (GE Healthcare, Waukesha, WI) with 8 CHC (manufacturer not specified) acquired in vivo MEGA-PRESS spectra in the ACC (scan parameters: TE = 68 ms; TR = 1800 ms with 332 acquisition averages, voxel size = 24 mL) in their study investigating the reproducibility of B2B scans [39]. The reproducibility results were 14.8 for CV%, while the ICC was only 0.16 [39]. However, given that 8 rather than 32 CHC was used and given a smaller sample size (*n* = 13) in the study by Mikkelsen et al., as opposed to *n* = 28 for Day 1 and *n* = 29 for Day 2 in our study, it is not surprising that better reproducibility parameters were observed in our study. Namely, 32 CHC produces better SNR gains by a factor of two compared to 8 CHC [31], while a larger sample size produces better reproducibility results [40].

The examination of Bland–Altman plots in our study indicated good agreement between B2B scans on both days in the dmPFC/ACC and dlPFC. We further assessed each GABA+ data point that fell above or below the limits of the agreement (i.e., points that were above or below 95% confidence interval) by investigating Day 1 and Day 2 B2B Bland–Altman plots for each quality parameter and conducting the assessment of quality-parameter outliers (points that are ± 1.96 SD above or below the mean) for each scan. The quality parameters investigated were GABA+ FWHM (frequency width at half maximum), H_2_O FWHM, GABA SNR, H_2_O SNR, GABA Fit Error %, and H_2_O Fit Error %. We briefly describe each quality parameter and its effect on GABA+ measurements below.

FWHM is the width of a metabolite curve measured at half maximum in ppm or Hz. It describes spectral resolution [41,42]. Due to the inhomogeneities of the scanner’s magnetic field, poor shimming procedures, or eddy currents leading to variations in resonant frequencies, FWHM widens, resulting in poorer metabolite peak separation, which affects the accuracy of metabolite quantification [41,42]. SNR is a measurement of an MRS signal divided by a standard deviation of the noise [42]. Given that SNR is generally affected by the magnetic field of the scanner, the inhomogeneity of the magnetic field and poor shimming techniques could lead to suboptimal SNR levels, which can lead to inaccurate metabolite measurements [27,41]. Fit error % indicates the deviations between the peak produced by the actual data compared to the modeled peak, which software uses to quantify metabolite concentrations [38,41]. Gannet 3.1 calculates fit error % by dividing the standard deviation of a residual peak (determined by the difference between the actual and modeled peaks) by the amplitude of the fitted peak [38]. Thus, the higher the difference between the actual and modeled peaks, the greater the fit error %, leading to less accuracy in metabolite concentrations. Considering that GABA+ concentrations were determined in references to unsuppressed H_2_O, FWHM, SNR, and fit error % for both GABA+ and H_2_O were evaluated. The means and standard deviations of quality parameters are shown in Appendix A for the dmPFC/ACC and Appendix A for the dlPFC.

The examination of Bland–Altman plots of GABA FWHM revealed that significant variation in GABA FWHM between B2B scans may have led to points a (Figure 2a), c (Figure 2b), d (Figure 2c), and f (Figure 2d) appearing outside the 95% confidence interval (CI). Moreover, Bland–Altman plots for point d H_2_O SNR demonstrated significant variations, and scan 2 GABA SNR for point d was 1.96 SD below the mean. Thus, deviations in GABA FWHM, H_2_O SNR, and GABA SNR may have additively affected the positioning of point d outside the 95% CI. Similarly, GABA SNR belonging to point e was 1.96 SD above the mean for scan 1, whereas point g GABA fit error% for scans 1 and 2 were 1.96 SD above their respective means. Here, too, the quality parameter deviations for points e and g may have contributed to their locations outside the 95% CI.

Removing Bland–Altman plot GABA+ concentration outliers displayed in Figure 2 improved the reproducibility of Day 1 and Day 2 B2B parameters. Thus, for the dmPFC/ACC, after removing the outliers, Day 1 B2B reproducibility values were CV% of 4.48, r = 0.84 (*p* < 0.001), and ICC = 0.83 (*p* < 0.0001), whereas for Day 2, they were CV% of 4.34, r = 0.83 (*p* < 0.001), and ICC = 0.83 (*p* < 0.0001). Similarly, removing Bland–Altman plot outliers in dlPFC increased Day 1 B2B reproducibility to CV% of 4.11, r = 0.79 (*p* < 0.001), and ICC = 0.77 (*p* < 0.0001) and Day 2 B2B to CV% of 5.80, r = 0.70 (*p* < 0.001), and ICC = 0.66 (*p* < 0.0001).

In summary, our spectral parameters demonstrated very good B2B reproducibility for GABA+ concentration in the dmPFC/ACC and dlPFC, and these parameters were comparable to those shown in similar studies. Considering that parameters determining spectral quality affected the reproducibility of B2B GABA+ concentration values, closer inspection of GABA+ quality parameters is warranted when measuring GABA+ concentration in the clinical setting and research studies.

### 3.2. D2D Reproducibility

To account for daily physiological variations in GABA+ concentrations and technical differences, we compared the D2D reproducibility of 5.2 min scans. The inspection revealed variations in CV% (5.44–7.08%) for the dmPFC/ACC and 6.85–8.15% for the dlPFC. Comparisons of the dlPFC generated relatively moderate reliability (ICC), 0.44–0.54, whereas, for the dmPFC/ACC, more drastic variations (poor to good; 0.32–0.70) in reliability were observed. Similarly, Pearson’s correlation assessments for the dlPFC were medium and ranged from 0.44 to 0.54 for the dlPFC, while for the dmPFC/ACC varied more drastically (0.33–0.71). A comparison of GABA+ concentrations in the dmPFC/ACC showed no correlation between S1D1 and S2D2 scans (*p* = 0.089). Next, we examined the effects of the degree of voxel overlaps on D2D reproducibility values; however, no statistically significant correlations were observed (*p* > 0.05) in either region. Inspecting voxel compositions for GM/WM/CSF between the assessment days revealed no differences (Table 6). Voxel tissue composition comparison was necessary, knowing that GABA+ concentrations were generally 2-fold higher in GM than WM [43,44], and these tissue D2D differences might have affected GABA+ concentration measurements and, subsequently, the D2D reproducibility.

Since the lower-than-expected D2D reproducibility of 5.2 min scans was noted, we next decided to average GABA+ concentrations for two scans within each daily session and compared D2D reproducibility between the daily averaged GABA+ values. By doing so, we assumed that averaged GABA+ concentrations represented a sum of acquisition averages (192 for scan 1 + 192 for scan 2 = 384 for averaged data) and total scan time (5.2 min + 5.2 min = 10.4 min for averaged GABA+ values). We expected that SNR would also increase by increasing the acquisition averages, given that SNR is directly proportional to the square root of the acquisition averages [27].

D2D comparison of averaged GABA+ values generated better reproducibility parameters in both ROIs. The within-subject variability parameters were more robust (4.95 CV% for the dmPFC/ACC and 5.85 CV% for the dlPFC), while the reliability for the dmPFC/ACC was good (ICC = 0.61) and moderate-to-good for the dlPFC (ICC = 0.58). Moreover, Pearson’s correlation coefficients between daily averaged scans were large and statistically significant in both ROIs.

Our reproducibility parameters were comparable to two other published studies [36,37]. Brix and colleagues examined D2D within-subject variability of GABA+ concentrations in the ACC (21 mL voxel size) [36]. In this study, 21 male participants were scanned one week apart, using a 3T GE Discover 750 MRI scanner (Milwaukee, WI) with 32 CHC (manufacturer not specified) and MEGA-PRESS with TE = 68 ms, TR = 1800 ms, 218 acquisition averages, and 4096 acquisition points [36]. The results showed a CV% of 6–13%, somewhat lower than our averaged dmPFC/ACC D2D results (CV% of 4.95%). However, considering that Brix and colleagues used a slightly smaller ACC voxel size, fewer acquisition averages, and a smaller sample size than in our study, CV% differences are comparable.

In contrast, Greenhouse and colleagues examined the D2D reproducibility of GABA+ by assessing a 25 mL dlPFC voxel. In this study, 20 male participants were scanned 16 ± 3 days apart using a 3T Siemens TIM/trio MRI scanner (Berlin/Munich, Germany) with the 32 CHC (manufacturer unspecified) and MEGA-PRESS sequence with TE = 68 ms, TR = 1500 ms, and 320 acquisition averages [37]. Within-subject repeatability was 4.6%, while reliability was excellent (ICC = 0.77). However, since Greenhouse and colleagues assessed a much larger dlPFC voxel (25 mL while ours was 18 mL) [37], which was expected to produce better SNR [32], more robust reproducibility parameters in this study compared to ours were not surprising.

Our study’s Bland–Altman plot analysis of D2D averaged GABA+ concentrations indicated excellent agreement in the dmPFC/ACC and dlPFC. In reference to their quality parameters, the analysis of Bland–Altman outliers revealed that significant GABA SNR variations occurred between averaged Day 1 and Day 2 GABA+ concentrations, as displayed by point a in the dmPFC/ACC (Figure 5a). Moreover, marginally significant variations in GABA SNR and H_2_O SNR were also observed between averaged daily GABA+ values in the dlPFC for point c (Figure 5b). For point d (Figure 5b), GABA SNR for Day 1 was 1.96 SD below the mean. Conversely, daily physiological differences may have influenced the positioning of point b outside the 95% CI (Figure 5a).

Like with B2B data, removing Bland–Altman outliers (Figure 5) generated better reproducibility parameters for averaged D2D comparisons of GABA+ concentrations in the dmPFC/ACC and dlPFC. Accordingly, the reproducibility parameters were 4.29 CV% and r = 0.73 (*p* < 0.0001); ICC = 0.72 (*p* < 0.0001) for the dmPFC/ACC, and 4.36 CV%, r = 0.80 (*p* < 0.0001), and ICC = 0.80 (*p* < 0.0001) for the dlPFC.

Overall, our D2D reproducibility data for GABA+ indicated that a higher number of acquisition averages than 192 with a longer scan time than 5.2 min (leading to better SNR) is needed to overcome the daily physiological, technical, and procedural variations in the dmPFC/ACC and dlPFC. More robust reproducibility was demonstrated by averaging two daily scans assuming that averaging produced GABA+ concentrations equivalent to those acquired with 384 averages. Our averaged GABA+ reproducibility parameters were comparable to those reported in other studies. Like with the B2B reproducibility data, several quality parameters apparently influenced the D2D reproducibility of GABA+ concentrations in the dmPFC/ACC and dlPFC, warranting more thorough investigations of these parameters in clinical/research settings.

### 3.3. Sex Differences

Previous research examining sex differences in GABA+ concentrations showed mixed results. In the study by O’Gorman and colleagues, investigating seven male and seven female participants (25–38 years of age) in the dlPFC, higher GABA+ levels were observed in male subjects [11]. Contrary to these findings, another investigation of GABA+ levels in the dlPFC comparing 19 males and 19 females (19–20 years of age) showed no statistically significant differences [45]. In a study in which a sex comparison was made in the ACC between 17 male and 24 female participants (34.8 ± 10 years of age), no differences concerning GABA+ concentrations were also detected [46]. Lastly, Gao and colleagues examined GABA+ in larger ROIs (frontal and parietal regions) in 51 female and 49 male participants (20–76 years of age) and found no differences in GABA+ between sexes [10].

We used GABA+ averaged data (which produced better D2D reproducibility) to compare sex differences on two separate days (Day 1 and Day 2) and found no differences in GABA+ in either the dlPFC or dmPFC/ACC. Since our study and studies by Grachev et al., Aufhaus et al., and Gao et al. [10,45,46] used larger sample sizes than O’Gorman et al. [11], it is possible that better statistical power produced more accurate results. Thus, it is reasonable to assume that the accumulated evidence from the abovementioned studies points to no significant differences in GABA+ between the sexes.

Moreover, we examined within-sex differences between two assessment sessions (Day 1 and Day 2), conducted 1–3 days apart (at approximately the same time), and found no variations in GABA+. The purpose of this comparison was to track sex-specific daily physiological variations. We were particularly interested in observing within-subject changes in females, considering that GABA+ concentrations were reported to be higher in the follicular than in the luteal phase of the menstrual cycle [47]. However, it is possible that in our study, approximately half of the female subjects were in the follicular and half in the luteal phase on each assessment day, leading to similar, averaged-out effects of the menstrual cycles on GABA+ concentrations. Thus, no differences in GABA+ concentrations were observed in within-female subjects.

### 3.4. Limitations

Several limitations were identified in our study. First, we relied on self-reports for assessments. No outcome measures or laboratory tests were used to confirm that our group of participants was without psychiatric/neurological conditions or that they were not taking any medications that would have potentially affected GABA+ concentrations in the dmPFC/ACC and dlPFC. Second, we did not track any daily dietary differences within or between our participants or the menstrual cycle differences for females [12,13,14,15,47]. As indicated in the Introduction, these physiological variations may have been able to explain daily variations in GABA+ concentrations. Third, our presented results are for a small sample of participants with a limited age range. Fourth, D2D reproducibility was assessed for scans conducted 1–3 days apart, which is not the reassessment time frame that generally occurs in clinical or research settings. For example, in longitudinal clinical trials, patients are reassessed weekly, bi-weekly, or monthly; thus, future studies should assess the D2D reproducibility of GABA+ concentrations during these time frames (as more significant physiological variations may be observed with larger reassessment time frames). Fifth, after observing the lower-than-expected D2D reproducibility of single 5.2 min scans (192 acquisition averages), we averaged GABA+ concentrations produced by two B2B scans to determine D2D reproducibility of the equivalent to 384 acquisition averages (10.4 min). Although good reproducibility was observed with the averaged scans, robust D2D reproducibility results could also have been obtained with fewer acquisition averages than 384. Thus, future studies should investigate the D2D reproducibility of data obtained between 192 and 384 acquisition averages. These data could be obtained by gradually merging spectral transients [36] between two scans on each assessment day. Despite these limitations, our study gives a good insight into the B2B and D2D reproducibility of GABA+ concentrations in the dmPFC and dlPFC at the set sequence parameters.

## 4. Materials and Methods

### 4.1. Participants

Twenty-nine healthy volunteers (mean age 23.9 ± 3.3 years; range, 20–32 years; and 15 female and 14 male) completed the scanning procedures for dmPFC/ACC, and 28 (mean age 24.0 ± 3.3; range, 20–32 years; and 15 female and 13 male) for dlPFC at the Centre for Addiction and Mental Health (CAMH). The recruitment was conducted by word of mouth. Before study enrollment, informed consent was obtained from all participants. Only male and female participants, 18 years and older, who have never been diagnosed with psychiatric and neurological illness, including significant brain trauma, migraines, and learning disabilities, were included in this study. For safety reasons, participants who were pregnant, breastfeeding, and those with metal or electronic implanted devices and severe claustrophobia were excluded from this study. Furthermore, all participants were asked to limit their caffeine intake to <200 mg/day (approximately amounting to no more than one cup of filtered coffee or two cups of instant coffee or tea), avoid drinking alcohol, and keep their general activities consistent both days. This study was approved by CAMH Research Ethics Board (REB) before initiating recruitment.

### 4.2. Scanning Procedures

The study participants were scanned four times using a 3T General Electric (GE) MR 750 scanner (General Electric, Waukesha, WI, USA) equipped with a 32-channel head coil (Nova Medical Inc., Wilmington, MA, USA). First, two scans were conducted B2B in a single session. Another set of B2B scans was conducted one to three days later at approximately a similar time.

High-resolution T1 weighted images were acquired using 3D IR fast spoil gradient (FSPGR) sequence BRAVO (TE = 3.0 ms, TR = 6.7 ms, inversion time (TI) = 650 ms, flip angle = 8°, resolution = 0.9 mm^3^, and scan time = 5 min). CHESS (Chemical Shift Selective Saturation) was used for water suppression [48]. MRS spectra were obtained from two ROIs; the dmPFC/ACC and dlPFC. The dmPFC/ACC voxel was placed on a reformatted oblique axial image parallel to the ACC and the corpus callosum on the sagittal plane. The voxel was kept away from the corpus callosum, skull, anterior frontal lobe, and peri-genual cingulate cortex, as demonstrated in Figure 7a. The voxel size was 4 cm (anterior-posterior) × 2 cm (right-left) × 3 cm (superior-posterior), 24 cm^3^. The dlPFC voxel was placed on a double oblique image between the superior and inferior frontal gyrus. dlPFC voxel size was 18 cm^3^ or 3 cm (anterior-posterior) × 3 cm (right-left) × 2 cm (superior-posterior), as seen in Figure 7b.

MEGA-PRESS [30] was used to acquire 1H MRS data. The sequence parameters for each voxel are outlined in Table 7. MEGA-PRESS uses J-difference editing, which acquires spectra under two different conditions (editing-ON and editing-OFF) throughout a scan in an interleaved manner. The editing pulses are radio frequency (RF) pulses with a pulse width = 14.4 ms. During the editing-ON acquisition, an editing pulse is placed at 1.9 ppm to invert GABA-H3 spins which refocuses the evolution of J-coupled GABA-H4 spins at 3.0 ppm. Conversely, during the editing-OFF acquisition, the editing pulse is placed at 7.5 ppm: a region with no metabolite signatures. The difference spectrum results from subtracting the two spectral acquisitions, which results in an uncovered GABA peak at 3.0 ppm [8,30,31,38]. However, this peak also contains an MM signal. The MM resonance peak at 1.7 ppm is near the editing-ON pulse and is coedited with GABA, producing a peak at 3.0 ppm in the difference spectrum. Given that the GABA-edited peak is contaminated with MM, we refer to the measurement of GABA as GABA+ (GABA + MM).

Water-unsuppressed reference spectra used for internal tissue/water referencing were acquired before the water-suppressed averages. AUTOSHIM (manufacturer’s automated shimming) procedures were conducted before each MRS scan to ensure the full width at half maximum (FWHM) ≤ 12.

### 4.3. MRS Data Processing

MEGA-PRESS data were processed and analyzed using Gannet 3.1. [38] Voxel-to-T1 weighted image co-registration was performed using Gannet 3.1 and SPM12 (www.fil.ion.ucl.ac.uk/spm, accessed on 3 January 2022). GABA+ values were reported in institutional units (I.U.). Acquired unsuppressed water was used as the internal reference. The results were corrected for water relaxation and density in the tissue compartments [49]. The correction factors were determined by FSL (FMRIB Software Library). The percentage of voxel overlaps between D2D scans was also assessed using FSL. Figure 8 shows Gannet 3.1 outputs with GABA+ peaks for a) dmPFC/ACC and b) dlPFC.

Visual inspection of the output spectra and fits was performed to ensure that only good-quality data were included in the analysis [27]. Accordingly, data points with noisy outputs, poor pre-post-GABA+ or Cr frequency alignments, high amounts of residual water, and prominent lipid peaks were eliminated from the analysis [27,38]. Furthermore, the data points with quality parameters (GABA+ FWHM, H_2_O FWHM, GABA+ fit error %, H_2_O fit error %, and ±3 SD away from the mean) were excluded from the study [50].

### 4.4. Statistical Analysis

SPSS version 28 (IBM, Chicago, IL, USA) was used for the statistical analysis. GABA+ values were displayed in terms of M ± SD. The reproducibility of B2B and D2D scans was assessed using Pearson’s correlation r, percentages of coefficient of variation (CV%), and the intra-class correlation coefficient (ICC). B2B and D2D within-sex differences were determined using the dependent t-test, while an independent *t*-test was used for assessing GABA+ concentration differences between males and females.

Pearson’s correlation r measures linear correlation. It measures the strength and directions of correlation. The power of association was ranked as either very small (0 < r < 0.1), small (0.1 < r < 0.29), medium (0.3 < r < 0.49), or large (0.5 < r < 1.0) [51]. CV% is a measure of repeatability, indicating within-subject variance. It was calculated by (M/SD) ×100%, where M represented the average of a subject’s two test–retest scans [52]. Conversely, ICC assessed reliability that considers both within- and between-subjects variance. In this analysis, SPSS was set to consider the single-rating, absolute agreement for two-way mixed effects ICC [53]. The quality of ICC parameters was determined according to the following convention: poor (ICC < 0.4), moderate (0.4 < ICC < 0.59), good (0.6 < ICC < 0.74), and excellent (ICC > 0.75) [52]. Moreover, the range of agreement between two test–retest scans (e.g., B2B or D2D) was visualized with Bland–Altman plots [54]. Quality parameters (GABA+ and H_2_O FWHM, GABA+ and H_2_O SNRs, GABA+ and H_2_O fit error %) were further examined for Bland–Altman GABA+ outliers to assess their effect on these data. The effects of differences in voxel overlaps and tissue heterogeneity on D2D reproducibility were evaluated by computing the correlation between the D2D percentage of voxel overlaps and the D2D reproducibility matrix.

Given that both participants’ physiological differences and scanner stability parameters may impact the evaluation of the reproducibility of 5.2 min D2D scans, D2D evaluation of two averaged B2B scans was also performed (as these scans would be considered twice as long: 2 × 5.2 min = 10.4 min) in case of lower-than-expected reproducibility.

## 5. Conclusions

In conclusion, the ^1^H MRS parameters used in our study produced excellent B2B repeatability and reliability of GABA+ in the dmPFC/ACC and dlPFC when using 192 acquisition averages with our new 32-channel head coil. The scan time was optimal to counteract any spectroscopic technical variation observed between B2B scans while minimizing the participant’s physiological changes. Moreover, excellent D2D reproducibility of averaged GABA+ concentrations was also found in both ROIs. Considering that larger procedural (adjusting voxel placement), technical, and physiological changes might have occurred when comparing the D2D reproducibility of GABA+, we compared the averaged GABA+ concentrations. With the averaging procedure, the assumption was made that number of acquisition averages was doubled (from 192 to 384) and that by producing better SNRs, we would obtain more robust D2D reproducibility results. Our findings were specific to the Gannet 3.1 GABA+ fitting software, which was selected for processing and quantifying GABA+ measurements because of its user-friendly interphase.

Our study demonstrated excellent GABA+ repeatability and reliability in the dmPFC/ACC and dlPFC. Further examination of GABA+ outliers could be helpful to enhance ^1^H MRS determined by Bland–Altman plots, which revealed that spectral quality parameters, such as FWHM, SNRs, and fit error %, may have affected the B2B and D2D reproducibility in both ROIs. Thus, a more thorough examination of these parameters may be necessary when measuring GABA+ in clinical/research settings. GABA+ concentration differences between and within male and female participants were less significant. Henceforth, when conducting clinical studies in which GABA+ is measured in the dmPFC/ACC and dlPFC, larger sample sizes may have to be used to neutralize any ‘false positive’ sex effects on GABA+ values. Overall, our ^1^H MRS parameters demonstrated good reproducibility of GABA+ results in the dmPFC/ACC and dlPFC, and these parameters could be used in future in vivo studies.

## Figures and Tables

**Figure 1 ijms-24-07713-f001:**
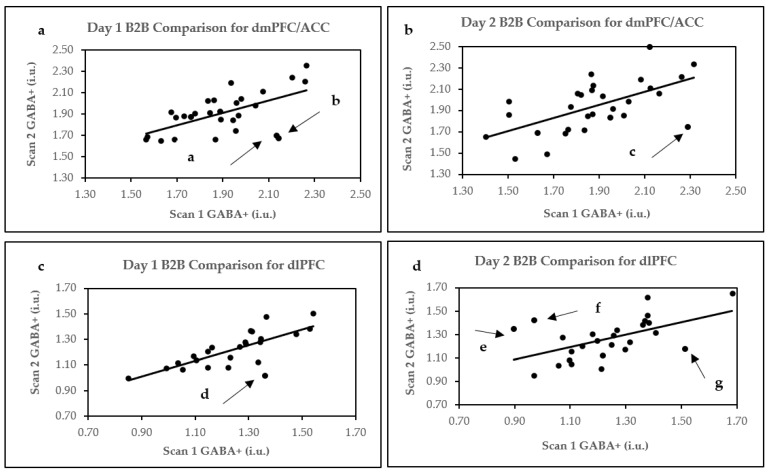
Pearson’s correlation plots for B2B comparisons. (**a**) Day 1 B2B in dmPFC/ACC; (**b**) Day 2 B2B in dmPFC/ACC; (**c**) Day 1 B2B in dlPFC; and (**d**) Day 2 B2B in dlPFC. Arrows and letters a–g indicate data points above or below 95% confidence intervals, as demonstrated by Bland–Altman plots.

**Figure 2 ijms-24-07713-f002:**
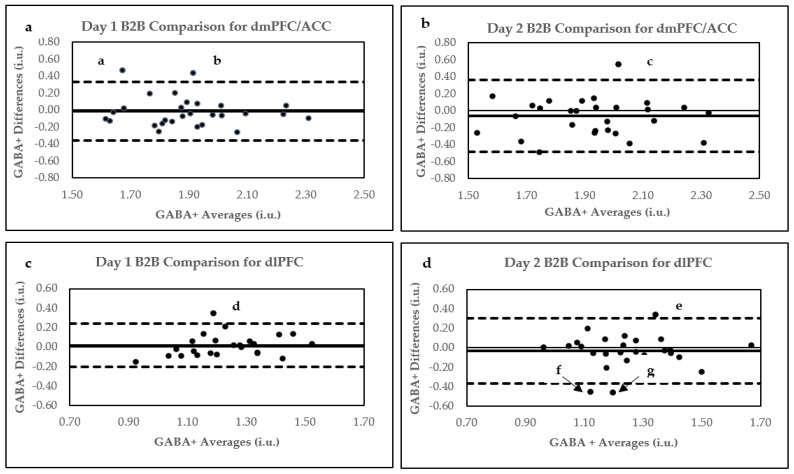
Bland–Altman plots for B2B comparisons. (**a**) Day 1 B2B in dmPFC/ACC (**b**) Day 2 B2B in dmPFC/ACC, (**c**) Day 1 B2B in dlPFC, and (**d**) Day 2 B2B in dlPFC. Letters a–g indicate data points outside upper and lower limits. Bland–Altman plots show agreement between two scans conducted within each scanning session. Within these plots, the solid line represents the mean difference between two methods of measurement (bias), and the broken upper lines indicate the upper limits of agreement (bias + 1.96 SD). In contrast, the lower broken lines indicate lower limits of agreement (bias − 1.96 SD).

**Figure 3 ijms-24-07713-f003:**
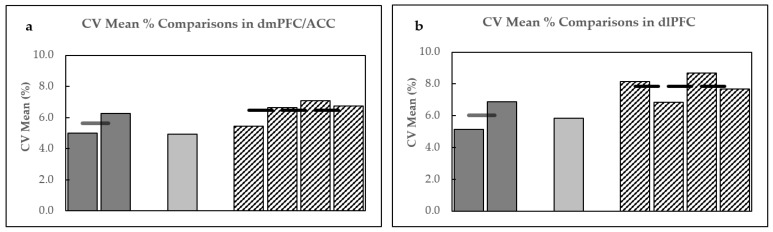
Within-subject CV% of B2B and D2D comparisons for GABA+ in (**a**) dmPFC and (**b**) dlPFC. Dark gray bars indicate Day 1 and Day 2 B2B measurements. Light gray bar represents averaged D2D measurements, while bars with black and white stripes indicate 5.2 min scan D2D measurements (from left to right, S1D1 and S1D2; S2D1 and S2D2; S1D1 and S2D2; and S2D1 and S1D2). Horizontal lines indicate the means of each set of scans indicated by bar chart clusters. CV%, coefficient of variation percentage; S, scan, D, day.

**Figure 4 ijms-24-07713-f004:**
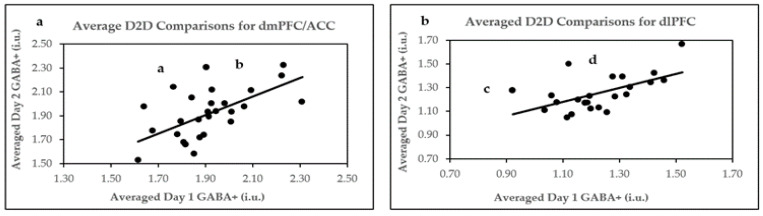
Pearson’s correlation plots for averaged D2D comparisons. (**a**) D2D averages in dmPFC/ACC; (**b**) D2D averages in dlPFC. Arrows and letters indicate data points above or below 95% confidence intervals, as demonstrated by Bland–Altman plots.

**Figure 5 ijms-24-07713-f005:**
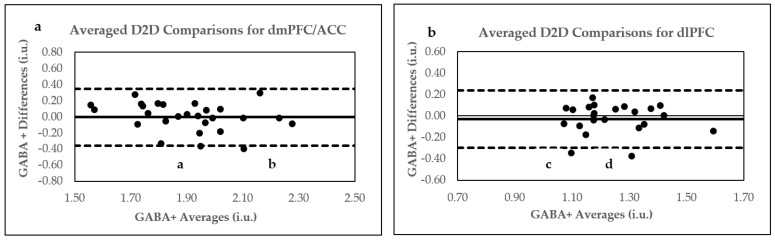
Bland–Altman plots for averaged D2D comparisons. (**a**) Averaged D2D in dmPFC/ACC (**b**) Averaged D2D in dlPFC. Letters a, b, c, and d indicate data points outside upper and lower limits. Bland–Altman plots show agreement between two scans conducted within each scanning session. Within these plots, the solid line represents the mean values of the difference dataset (bias), and the broken upper lines indicate the upper limits (bias + 1.96 SD). In contrast, the lower broken lines indicate lower limits (bias − 1.96 SD).

**Figure 6 ijms-24-07713-f006:**
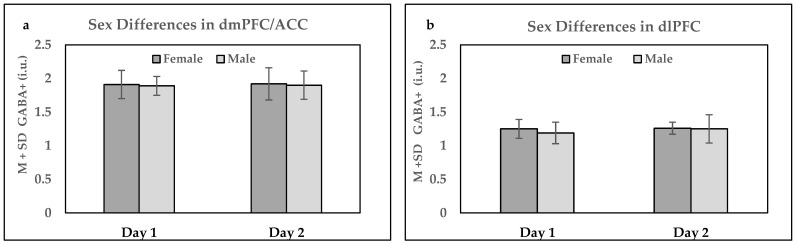
Within- and between-subject GABA+ sex differences in (**a**) dmPFC/ACC and (**b**) dlPFC. Dark gray bars indicate the average GABA+ concentration for females for Day 1 and Day 2. The light gray bar represents mean GABA+ concentration differences for males. M mean, SD, standard deviations.

**Figure 7 ijms-24-07713-f007:**
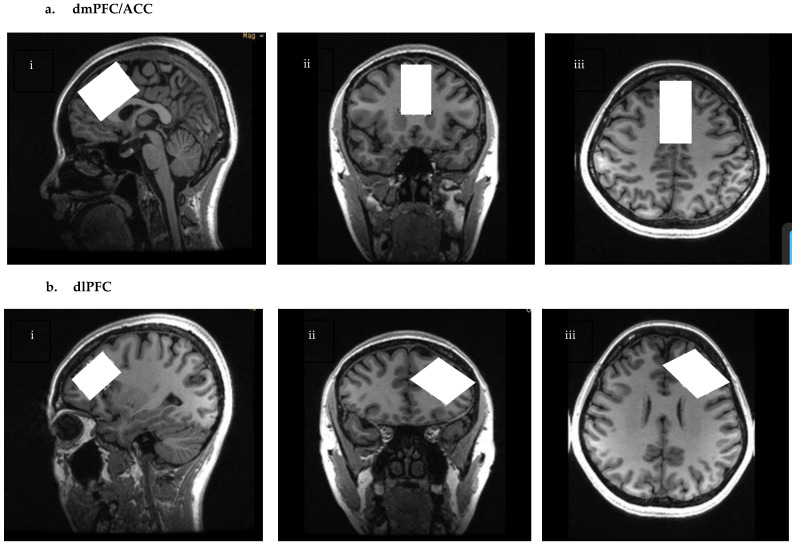
Images illustrating voxel placement for (**a**) dmPFC/ACC and (**b**) dlPFC. For dmPFC/ACC, the voxel is placed 4 cm in the anterior-posterior, 2 cm right-left, and 2 cm in the superior-inferior direction (with 24 cm^3^ total volume). For dlPFC, the placement is 3 cm in the anterior-posterior, 3 cm right-left, and 2 cm in the superior-inferior direction (with 18 cm^3^ total volume). The voxel placement is presented in (**i**) sagittal, (**ii**) axial, and (**iii**) coronal views. The white rectangles indicate voxel placements.

**Figure 8 ijms-24-07713-f008:**
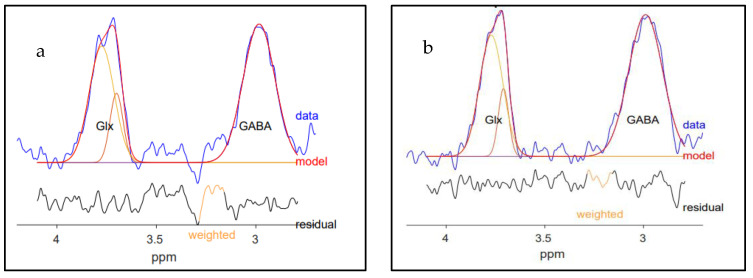
Modeling of GABA+ signal in (**a**) dmPFC/ACC and (**b**) dlPFC. The GABA+ -edited spectrum acquired from GABA+ measurements is shown in blue, while the modeled curve is red. The black lines demonstrate the residual between blue and red lines. Gannet software uses a simple Gaussian model by default.

**Table 1 ijms-24-07713-t001:** B2B reproducibility parameters for dmPFC/ACC-*n*, a number of participants; CV%, within-subject coefficient of variation; ICC, inter-class correlation coefficient.

Parameters	Day 1 B2B	Day 2 B2B
*n*	28	29
Pearson’s correlation r	0.59 ***	0.60 ***
Within-subject mean CV%	5.02	6.26
ICC	0.60 ***	0.59 ***

Correlation is significant at *** 0.001 (2-tailed).

**Table 2 ijms-24-07713-t002:** B2B reproducibility parameters for dlPFC. *n*, number of participants; CV%, within-subject coefficient of variation; ICC, inter-class correlation coefficient.

Parameters	Day 1 B2B	Day 2 B2B
*n*	24	25
Pearson’s correlation r	0.74 ***	0.54 **
Within-subject mean CV%	5.15	6.89
ICC	0.73 ***	0.54 **

Correlation is significant at ** 0.01; *** 0.001 (2-tailed).

**Table 3 ijms-24-07713-t003:** D2D reproducibility parameters for dmPFC/ACC. *n*, number of participants; CV%, within-subject coefficient of variation; ICC, inter-class correlation coefficient; S, scan; D, day.

Parameters	S1D1 and S1D2	S2D1 and S2D2	S1D1 and S2D2	S1D2 and S2D1
*n*	28	29	28	29
Pearson’s correlation r	0.71 ***	0.37 *	0.33	0.57 **
Within-subject mean CV%	5.44	6.63	7.08	6.73
ICC	0.70 ***	0.36 *	0.32 *	0.56 **

Correlation is significant at * 0.05 level; ** 0.01; *** 0.001 (2-tailed).

**Table 4 ijms-24-07713-t004:** D2D reproducibility parameters for dlPFC. *n*, number of participants; CV%, within-subject coefficient of variation; ICC, inter-class correlation coefficient; S, scan; D, day.

Parameters	S1D1 and S1D2	S2D1 and S2D2	S1D1 and S2D2	S1D2 and S2D1
*n*	24	24	24	24
Pearson’s correlation r	0.53 **	0.52 **	0.44 *	0.53 **
Within-subject mean CV%	8.15	6.85	8.68	7.69
ICC	0.54 **	0.49 **	0.44 *	0.52 **

Correlation is significant at * 0.05 level; ** 0.01; (2-tailed).

**Table 5 ijms-24-07713-t005:** Averaged D2D reproducibility parameters for dmPFC/ACC and dlPFC. *n*, number of participants; CV%, within-subject coefficient of variation; ICC, inter-class correlation coefficient.

Parameters	dmPFC/ACC D2D	dlPFC D2D
*n*	28	23
Pearson’s correlation r	0.62 ***	0.58 **
Within-subject mean CV%	4.95	5.85
ICC	0.61 ***	0.58 **

Correlation is significant at ** 0.01; *** 0.001 (2-tailed).

**Table 6 ijms-24-07713-t006:** Tissue composition fractions of GM/WM/CSF for dmPFC/ACC and dlPFC. Average indicates the mean of Day 1 and Day 2 values. ROI, region of interest; *n*, number of participants; GM, gray matter; WM, white matter; CSF, cerebrospinal fluid; M, mean; SD, standard deviation.

ROI	Tissue Fractions (M ± SD)	Day 1	Day 2	Average
dmPFC/ACC (*n* = 28)	GM	0.62 ± 0.03	0.62 ± 0.03	0.62 ± 0.03
	WM	0.15 ± 0.03	0.15 ± 0.03	0.15 ± 0.03
	CSF	0.23 ± 0.05	0.23 ± 0.05	0.23 ± 0.05
dlPFC (*n* = 23)	GM	0.48 ± 0.04	0.49 ± 0.03	0.49 ± 0.03
	WM	0.45 ± 0.06	0.45 ± 0.04	0.45 ± 0.04
	CSF	0.07 ± 0.02	0.07 ± 0.02	0.07 ± 0.02

**Table 7 ijms-24-07713-t007:** MEGA-PRESS sequence parameters for dmPFC/ACC and dlPFC. There were 96 editing-ON, 96 editing-OFF acquisitions, and 16 unsuppressed water acquisitions for both scans.

Sequence Parameters	dmPFC/ACC	dlPFC
Echo time (TE)	68 ms	68 ms
Repetition time (TR)	1500 ms	1500 ms
Number of acquisitions	192	192
Number of excitations (NEX)	8	8
Number of points	4096	4096
Spectral width	5000 Hz	5000 Hz
Scan time	5.2 min	5.2 min

## Data Availability

Data sharing is not applicable.

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
