# Peer review of "Evaluating Back-to-Back and Day-to-Day Reproducibility of Cortical GABA+ Measurements Using Proton Magnetic Resonance Spectroscopy (1H MRS)"

_ijms, 2023, doi:10.3390/ijms24097713_

Round 1

Reviewer 1 Report

In the manuscript entitled "Evaluating Back-to-back and Day-to-day Reproducibility of Cortical GABA+ Measurements Using Proton Magnetic Resonance Spectroscopy (1H MRS)" proposes to assess the reproducibility of GABA measurements back-to-back and day-to-day in healthy participants. In this way, the authors propose to investigate daily physiological variations as well as technical stability. 
The introduction offers enough and relevant background (incl. references). 
The methods and results are comprehensively described. I identified no obvious problem or question. HOWEVER, and although the authors found no statistical significant differences between males and females (subtitle 2.5), it would be interesting to still include the graphs of these data! p values are not everything. 
Discussion, is also comprehensive and transparent (outliers situation was properly addressed). 
At the moment, most reports point towards a resemblance of GABA levels between sexes. Authors addressed succintly considerations on male -  female endocrinology / menstrual cycles phases. Indeed, to establish similar circunstances of hormonal status between males and females is not easy. And GABA is a known neurotransmitter also in the ovaries. I wonder if future studies can quantify different neurotransmitters, not only in the CNS, but also in the peripheral and sensory systems. 

I am not sure if IJMS is the best journal for this study but it is comprehensive and well-written, and with well-tested hypotheses, using only a Tesla MRI scanner with a 32 channel head coil and MEGA-PRESS. 
It is not novel, presents a few interesting twists and is a very specific study that will interest mostly radiologists/health imaging experts and CNS researchers. 

Author Response

Dear Reviewer 1,

Thank you so much for reviewing our article titled: “Evaluating Back-to-back and Day-to-day Reproducibility of Cortical GABA+ Measurements Using Proton Magnetic Resonance Spectroscopy (1H MRS)”.  We have drafted our responses to your suggestions and request below:

  1. The methods and results are comprehensively described. I identified no obvious problem or question. HOWEVER, and although the authors found no statistically significant differences between males and females (subtitle 2.5), it would be interesting to still include the graphs of these data! p values are not everything.

  • We have now created Figure 6 plotting GABA+ sex differences in a) dmPFC/ACC and b) dlPFC

  1. I wonder if future studies can quantify different neurotransmitters, not only in the CNS, but also in the peripheral and sensory systems.

  • It is our understanding that techniques such as liquid chromatography-tandem mass spectrometry could be used for measuring certain neurotransmitters from the peripheral systems. Liquid chromatography separates different neurotransmitters, while mass spectrometry provides spectral information that may help identify and quantify the amount of these neurotransmitters.

  1. I am not sure if IJMS is the best journal for this study but it is comprehensive and well-written, and with well-tested hypotheses, using only a Tesla MRI scanner with a 32 channel head coil and MEGA-PRESS. 

  • We selected to publish our study in the IJMS because we felt that the target audience was appropriate, as molecular scientists studying GABA+ in the human brain could consider using the 1H MRS MEGA-PRESS sequence with the parameters specified in our paper.

Thank you again for reviewing our manuscript.  Please do not hesitate to let us know should you need further clarification regarding our responses.

Sincere regards,

Sonja Elsaid on behalf of all authors

Reviewer 2 Report

The manuscript entitled "Evaluating Back-to-back and Day-to-day Reproducibility of Cortical GABA+ Measurements Using Proton Magnetic Resonance Spectroscopy (1H MRS)" is a very interesting article which utilizes the advance technology to answer research hypothesis related to the major inhibitory neurotransmitter changes. In this article, the authors studied in detail about the GABA level alterations from different brain regions including dorsolateral pre-frontal cortex/anterior cingulate cortex relevant to psychiatric patients.

The weakness in the article

Line 1-4: Consider whether the uppercase and lower case for the wording such as "Back-to-back", "Day-to-day" need to be consistent (may not require upper case for one "back" and lower case for another "back" and so on). 

Line 36: Please provide the concrete conclusions "Our data reliable and comparable to other similar studies". Please make the sense of data and summarize your results. Readers might not understand what those increase % value means. Are these increase statistically significant?

Line 56-Line 64: The introduction section is currently unable to tap broader perspective of defense system. I suggest the author to bring the context of Glutamate-GABA recycling and glutathione defense system into the context. For latest development in the field utilize below relevant paper.   

PMID: 35011559

Line 116-117: Table 1: I suggest moving the text "Day 1............Coefficient" text below the Table 1. However, it is ok if you want to provide generalized title in bold in the Table in a very short text (if needed). Line 112 already have some text that are repeated with your current title of the Table. I would rather avoid redundancy wherever possible and try to portray the main message what we actually see in the Table.

In addition, you can add the "*" to the value that are statistically significant and define at what level of significance the value was tested below the table and including the statistic used.

For rest of all the Table: Table 2-6. Please follow the above comments similar to Table 1 and move the text below the Table 1.

Table 6 or elsewhere: I also suggest you write short summary below each table what are the main findings portray in the Table itself (wherever possible).

Figures 1-5: Line 135-217: all the figure legends must be placed below the Figures. Text used (x-axis, y-axis, unit etc.) in the figures itself are blurred. 

use the same font, size, and consistency in all the figures.

Figure 1: please provide more details things like what that arrow indicates, what we are observing here.

Figure 3: What is those horizontal lines overlapping above the graph bar figure 3a and 3b? If you want to show the statistically significant align your graph presentation properly. This type of error must be fixed.

Figure 6: I suggest providing more detail information about how this image was obtained (such as T1, T2 or what instrument used to obtain these figures, what intensity, power used for more details so that anyone can acquire same data without missing any details).

Figure 6, Table 7, Figure 7: I strongly believe that all the figures should be placed in the result section. If the figures are experimental set up, design, protocol paradigm then it can be given in the Methods section. Therefore, I suggest moving it in the appropriate location.

In addition, please define what that rectangle bar that are inserted in the image indicates in the figure legend. If possible, give different colors (such as yellow) so that readers will not have hard times to mix up whether the figures are experimentally obtained, or you have inserted it. 

I would also suggest the authors to discuss the GABA, Glutamate imbalance, and antioxidant system.

During work-place settings, the head injury or toxicants related event can trigger sequence of alteration in the transmitter system, astrogliosis, release of inflammatory cascade. You have not touched those parts either in the introduction or the discussion sections how those neurotransmitters.

For more relevant context, you could find more information from the below paper.

PMID:36768596

Author Response

Dear Reviewer 2,

Thank you so much for reviewing our article titled: “Evaluating Back-to-back and Day-to-day Reproducibility of Cortical GABA+ Measurements Using Proton Magnetic Resonance Spectroscopy (1H MRS)”.  We have drafted our responses to your suggestions and request below:

  1. Line 1-4: Consider whether the uppercase and lower case for the wording such as "Back-to-back", "Day-to-day" need to be consistent (may not require upper case for one "back" and lower case for another "back" and so on).

  • Please note that according to the IJMS template, every word in the article title starts with an uppercase letter, therefore “Back-to-back” and “ Day-to-day” appear as such

  1. Line 36: Please provide the concrete conclusions "Our data reliable and comparable to other similar studies". Please make the sense of data and summarize your results. Readers might not understand what those increase % value means. Are these increase statistically significant?
  • Please note that we have removed the sentence reading: “Our data was reliable and comparable to other similar studies” with the sentence reading: “Our study found excellent GABA+ repeatability and reliability in dmPFC/ACC and dlPFC.”

  • Considering that the article has 200-word limit, we were unable to include p-values for Pearson’s r correlations, and interclass-correlation coefficients; however, we have described the standard convention for these parameters in the Methods section on page 15, lines 1019 - 1028, as follows

“Pearson’s correlation r measures linear correlation. It measures the strength and directions of correlation. The power of association was ranked as either very small (0 < r < 0.1), small (0.1 < r < 0.29), medium (0.3 < r < 0.49), or large (0.5 < r < 1.0) [51]. CV% is a measure of repeatability, indicating within-subject variance. It was calculated by (M/SD) x 100%, where M represented the average of a subject’s two test-retest scans [52]. Conversely, ICC assessed reliability that considers both within- and between-subjects variance. In this analysis, SPSS was set to consider the single-rating, absolute agreement for two-way mixed effects ICC [53]. The quality of ICC parameters was determined according to the following convention: poor (ICC < 0.4), moderate (0.4 < ICC < 0.59), good (0.6 < ICC < 0.74), and excellent (ICC > 0.75) [52]

  1. Line 56-Line 64: The introduction section is currently unable to tap broader perspective of defense system. I suggest the author to bring the context of Glutamate-GABA recycling and glutathione defense system into the context. For latest development in the field utilize below relevant paper. PMID: 35011559
  • A detailed description of glutamate-GABA recycling and glutathione defense system are beyond the scope of this article; however, we have now revised our Introduction on pages 1-2, lines 43-62 to read:

“γ-aminobutyric acid (GABA) is a major inhibitory neurotransmitter in the mammalian brain. Together with glutamate and glutathione, GABA is linked to metabolic pathways in the brain [1,2]. GABA is synthesized through the decarboxylation of glutamate by two isozymes of glutamic acid decarboxylase (GAD) 65 and GAD 67 [3]. It acts through ionotropic GABAA and metabotropic GABAB receptors, involved in the neuromodulation of inhibitory and excitatory brain currents [4,5]. Aside from binding to GABA receptors at the synapse, GABA is transported to glial cells. Inside glial cells, it enters GABA shunt via GABA transaminase (GAT) directly into the Krebs cycle, the key metabolic pathway for cellular energy production [1,6]. From the Krebs cycle, glutamate is synthesized and converted into GABA, completing the energy supply cycle in the GABA shunt. Additionally, glutamate is one of the three amino acids needed to synthesize glutathione, which is essential for the degradation and synthesis of proteins and protects the cell against reactive oxygen species [1,6,7]. Due to the metabolic linkage, the concentrations between GABA, glutamate, and glutathione are related [1,6].”

  1. Line 116-117: Table 1: I suggest moving the text "Day 1............Coefficient" text below the Table 1. However, it is ok if you want to provide generalized title in bold in the Table in a very short text (if needed). Line 112 already have some text that are repeated with your current title of the Table. I would rather avoid redundancy wherever possible and try to portray the main message what we actually see in the Table.

  • Please note that according to the IJMS template, the table title and the legend should appear on top of the table.
  • We have now modified Table 1 title to read: “B2B reproducibility parameters for dmPFC/ACC”

  1. In addition, you can add the "*" to the value that are statistically significant and define at what level of significance the value was tested below the table and including the statistic used.
  • We have now removed the p-values from Tables 1-5, and added “*” to the statistically significant values. Each table now has a footnote that reads: “Correlation is significant at * 0.05 level; ** 0.01; *** 0.001 (2-tailed).”

  1. For rest of all the Table: Table 2-6. Please follow the above comments similar to Table 1 and move the text below the Table 1.

  • We have now modified Table 2 title to read: “B2B reproducibility parameters for dlPFC”
  • Please note that according to the IJMS template, the table title and the legend should appear on top of the table
  1. Table 6 or elsewhere: I also suggest you write short summary below each table what are the main findings portray in the Table itself (wherever possible).

  • Please note that according to the IJMS template, the table title and the legend should appear on top of the table
  • We have described each parameter presented in Table 6 in the table legend, which appears above the table and reads on page 7, lines 573-576, as follows:

“Table 6. Tissue composition (fractions of GM/WM/CSF adding to 1) for dmPFC/ACC and dlPFC. Average indicates the average of Day 1 and Day 2. ROI, region of interest; n, number of participants; GM, gray matter; WM, white matter; CSF, cerebrospinal fluid; M, mean; SD, standard deviation.”

  • Furthermore, the reason for including data presented in Table 6 is described in the Discussion section on page 10, lines 726-731.

“Inspecting voxel compositions for GM/WM/CSF between the assessment days revealed no differences (Table 6). Voxel tissue composition comparison was necessary, knowing that GABA+ concentrations were generally 2-fold higher in GM than WM [43,44], and these tissue D2D differences might have affected GABA+ concentration measurements and, subsequently, the D2D reproducibility.”

  1. Figures 1-5: Line 135-217: all the figure legends must be placed below the Figures. Text used (x-axis, y-axis, unit etc.) in the figures itself are blurred.

  • Please note that we have now moved the Figure legends below each figure and did our best to eliminate the blurriness. We have also submitted our figures separately, so that the IJMS publishing team could improve the article visibility in the publication-ready version.

  1. use the same font, size, and consistency in all the figures.

  • We have now modified all the figures so that they are all consistent in font and size.

  1. Figure 1: please provide more details things like what that arrow indicates, what we are observing here.
  • Please note that the arrows point to the data points indicated with letters a, b, c, d, e, f and g. The figure legend describes these points: “Letters a, b, c, d, e, f, and g indicate data points above or below 95% confidence intervals, as demonstrated by Bland-Altman plots.”

  1. Figure 3: What is those horizontal lines overlapping above the graph bar figure 3a and 3b? If you want to show the statistically significant align your graph presentation properly. This type of error must be fixed.
  • Please note that horizontal lines do not represent statistically significant data but bar cluster averages, as indicated in the Figure 3 legend: “Horizontal lines indicate the means of each set of scans indicated by bar chart clusters.”

  1. Figure 6: I suggest providing more detail information about how this image was obtained (such as T1, T2 or what instrument used to obtain these figures, what intensity, power used for more details so that anyone can acquire same data without missing any details).

  • Please note that the detailed information on how images were obtained is provided in the text, so that anyone wishing to use the same 1MRS MEGA-PRESS sequence parameters, and voxel sizes could do so. The information provided is on page 13, lines 882-917, as follows:

“4.2 Scanning Procedures

The study participants were scanned four times using a 3T GE MR 750 scanner (General Electric, Waukesha, WI, USA) equipped with a 32-channel head coil (Nova Medical Inc., Wilmington, MA, USA). First, two scans were conducted B2B in a single session. Another set of B2B scans was conducted one to three days later at approximately a similar time.

High-resolution T1 weighted images were acquired, using 3D IR fast spoil gradi-ent (FSPGR) sequence BRAVO (TE = 3.0 ms, TR = 6.7 ms, inversion time (TI) = 650 ms, flip angle = 8°, resolution = 0.9 mm3, scan time = 5 minutes). CHESS (Chemical Shift Selective Saturation) was used for water suppression [48]. MRS spectra were obtained from two ROIs; the dmPFC/ACC and dlPFC. The dmPFC/ACC voxel was placed on a reformatted oblique axial image parallel to the ACC and the corpus callosum on the sagittal plane. The voxel was kept away from the corpus callosum, skull, anterior frontal lobe, and peri-genual cingulate cortex, as demonstrated in Figure 7a. The voxel size was 4 cm (anterior-posterior) x 2 cm (right-left) x 3 cm (superior-posterior), 24 cm3. The dlPFC voxel was placed on a double oblique image between the superior and inferior frontal gyrus. dlPFC voxel size was 18 cm3 or 3 cm (anterior-posterior) x 3 cm (right-left) x 2 cm (superior-posterior), as seen in Figure 7b.

MEGA-PRESS [30] was used to acquire 1H MRS data. The sequence parameters for each voxel are outlined in Table 7. MEGA-PRESS uses J-difference editing, which ac-quires spectra under two different conditions (editing-ON and editing-OFF) through-out a scan in an interleaved manner. The editing pulses are radio frequency (RF) pulses with a pulse width = 14.4 ms. During the editing-ON acquisition, an editing pulse is placed at 1.9 ppm to invert GABA-H3 spins which refocuses the evolution of J-coupled GABA-H4 spins at 3.0 ppm. Conversely, during the editing-OFF acquisition, the edit-ing pulse is placed at 7.5 ppm: a region with no metabolite signatures. The difference spectrum results from subtracting the two spectral acquisitions, which results in an uncovered GABA peak at 3.0 ppm [8,30,31,38]. However, this peak also contains an MM signal. The MM resonance peak at 1.7 ppm is near the editing-ON pulse and is co-edited with GABA, producing a peak at 3.0 ppm in the difference spectrum. Given that the GABA-edited peak is contaminated with MM, we refer to the measurement of GABA as GABA+ (GABA + MM).

Water-unsuppressed reference spectra used for internal tissue/water referencing were acquired before the water-suppressed averages. AUTOSHIM (manufacturer’s automated shimming) procedures were conducted before each MRS scan to ensure the full width at half maximum (FWHM) ≤ 12..”

  1. Figure 6, Table 7, Figure 7: I strongly believe that all the figures should be placed in the result section. If the figures are experimental set up, design, protocol paradigm then it can be given in the Methods section. Therefore, I suggest moving it in the appropriate location.

  • Please note that Figure 6, Table 7 and Figure 7 support the experimental set up and study design and for that reason we have placed in the Methods section.

  1. In addition, please define what that rectangle bar that are inserted in the image indicates in the figure legend. If possible, give different colors (such as yellow) so that readers will not have hard times to mix up whether the figures are experimentally obtained, or you have inserted it.

  • To Figure 6, we have now added a sentence that reads: “The white rectangles indicate voxel placements.”
  • The article images and figures are to appear without the colour, however, voxel placements are now indicated with white rectangles.

  1. I would also suggest the authors to discuss the GABA, Glutamate imbalance, and antioxidant system.

  • A detailed description of glutamate-GABA recycling and glutathione defense system are beyond the scope of this article; however, we have now revised our Introduction on pages 1-2, lines 43-62 to read:

“γ-aminobutyric acid (GABA) is a major inhibitory neurotransmitter in the mammalian brain. Together with glutamate and glutathione, GABA is linked to metabolic pathways in the brain [1,2]. GABA is synthesized through the decarboxylation of glutamate by two isozymes of glutamic acid decarboxylase (GAD) 65 and GAD 67 [3]. It acts through ionotropic GABAA and metabotropic GABAB receptors, involved in the neuromodulation of inhibitory and excitatory brain currents [4,5]. Aside from binding to GABA receptors at the synapse, GABA is transported to glial cells. Inside glial cells, it enters GABA shunt via GABA transaminase (GAT) directly into the Krebs cycle, the key metabolic pathway for cellular energy production [1,6]. From the Krebs cycle, glutamate is synthesized and converted into GABA, completing the energy supply cycle in the GABA shunt. Additionally, glutamate is one of the three amino acids needed to synthesize glutathione, which is essential for the degradation and synthesis of proteins and protects the cell against reactive oxygen species [1,6,7]. Due to the metabolic linkage, the concentrations between GABA, glutamate, and glutathione are related [1,6].”

  1. During work-place settings, the head injury or toxicants related event can trigger sequence of alteration in the transmitter system, astrogliosis, release of inflammatory cascade. You have not touched those parts either in the introduction or the discussion sections how those neurotransmitters. For more relevant context, you could find more information from the below paper PMID:36768596

  • We have now included some relevant information in the Introduction section related to traumatic brain injury, although more detailed description of this topic is beyond the scope of this article. The included text is on page 2, lines 80-85:

“Following traumatic brain injury, disturbances in energy metabolism led to decreases in glutamate and increases in GABA and glutathione levels [2,25,26]. Since glutathione is synthesized from glutamate, depletions in glutamate may be due to increased glutathione synthesis to mitigate oxidative stress. Increased GABA may be a compensatory mechanism activated to reduce neuronal hyperactivity [2,25,26]. “

Reviewer 3 Report

The article is interesting, however, it looks rather "hermetic" and not easy to understand for people, who are not NMR spectroscopists.

A minor revision is needed, following points should be addressed:

1) The Authors should explain the significance of their study. What kind of information, not available before, is being provided ? How can the results be applied in medical diagnostics ? The statement that the results are in agreement with those obtained by other Authors is not very attractive. An up-grade of the Head coil in the Department is rather a questionable explanation of motivation to perform the study.

2) The Authors should read the text and go through the data thoroughly in order to show it in a way that is more comprehensible for non-specialists.

Author Response

Dear Reviewer 3,

Thank you so much for reviewing our article titled: “Evaluating Back-to-back and Day-to-day Reproducibility of Cortical GABA+ Measurements Using Proton Magnetic Resonance Spectroscopy (1H MRS)”.  We have drafted our responses to your suggestions and request below:

  • The Authors should explain the significance of their study. What kind of information, not available before, is being provided ? How can the results be applied in medical diagnostics ?

  • Please note that we modified the Introduction section on page 3 , lines 281 – 308 to highlight the significance of the study, how the study results could be applied to medical diagnostics and specify information not available prior to conducting our study:

“The best way to conduct these studies is to examine back-to-back (B2B) and day-to-day (D2D), test-retest reproducibility of GABA+. The purpose of B2B scan assessments is to assess the reproducibility of the technical variables (e.g., hardware, software-related, MRS sequence parameters) while attempting to keep the physiological variations minimal. The scan time and spectroscopic parameters determined in B2B studies could be used in patient single scan clinical assessment or case studies. The aim of comparing the reproducibility parameters of scans conducted on different days is to capture the effects of participants’ daily physiological variations and technical stability. Since dai-ly physiological variations are assumed in D2D reproducibility experiences, the scan time and spectroscopic parameters from these experiments could be used in clinical investigations necessitating repeated patient MRS assessments or cross-sectional or longitudinal clinical experiments.

Patients with psychiatric symptoms often experience discomfort if asked to undergo MRI scanning procedures for longer periods (more than 10 minutes) [34]; thus the general aim of MRS reproducibility studies is to find spectroscopic parameters utilizing the shortest possible scan time. Previous research demonstrated adequate B2B reproducibility with a 10-minute scan time, using the 3T scanner and 64-channel head coil (CHC) in anterior cingulate cortex (ACC) and dorsolateral prefrontal cortex (dlPFC) [35]. Moreover, D2D reproducibility with a 13- and 8.4- minutes scan times was shown with the 3T scanner and 32-CHC in the ACC and dlPFC, respectively [36,37].

With the growing demand for further reduction in the MRS scan time, our study aims to examine B2B and D2D test-retest reproducibility of GABA+ using the 3T scan-ner, 32 CHC and 5.2-minute scan time. In this study, the dorsomedial prefrontal cortex (dmPFC)/ACC and dlPFC regions-of-interest (ROI) will be scanned in healthy volun-teers (age range 20-32), and spectroscopy data will be processed with a user-friendly software, Gannet 3.1 [38]. Our exploratory analysis assessed within- and between-sex differences in GABA+ concentrations, considering mixed evidence in GABA+ concentrations between male and female participants [11].”

  • The statement that the results are in agreement with those obtained by other Authors is not very attractive. An up-grade of the Head coil in the Department is rather a questionable explanation of motivation to perform the study.
  • Please note that we have removed the sentence reading: “Our data was reliable and comparable to other similar studies” with the sentence reading: “Our study found excellent GABA+ repeatability and reliability in dmPFC/ACC and dlPFC” from the Abstract section.
  • Moreover, in the Conclusion, the sentence starting with: “ Our B2B and averaged D2D reproducibility parameters were comparable to those observed in other studies. “ was replaced by “Our study found excellent GABA+ repeatability and reliability in dmPFC/ACC and dlPFC.”

  • The Authors should read the text and thoroughly review the data to show it in a more comprehensible way for non-specialists.
  • Please note that we did our best to write this article in a comprehensible language for a general scientific audience. In fact, using the general language, we have described basic MRI parameters, such as SNR, FWHM, and  Fit error % in the Discussion section:

“FWHM is the width of a metabolite curve measured at half maximum in ppm or Hz. It describes spectral resolution [33,34]. Due to the inhomogeneities of the scanner’s magnetic field, poor shimming procedures, or eddy currents leading to variations in resonant frequencies, FWHM widens, resulting in poorer metabolite peak separation, which affects the accuracy of metabolite quantification [33,34]. SNR is a measurement of an MRS signal divided by a standard deviation of the noise [34]. Given that SNR is generally affected by the magnetic field of the scanner, the inhomogeneity of the magnetic field and poor shimming techniques could lead to suboptimal SNR levels, which can lead to inaccurate metabolite measurements [21,33]. Fit error % indicates the deviations between the peak produced by the actual data compared to the modeled peak, which software uses to quantify metabolite concentrations [29,33]. Gannet 3.1 calculates fit error % by dividing the standard deviation of a residual peak (determined by the difference between the actual and modeled peaks) by the amplitude of the fitted peak [29]. Thus, the higher the difference between the actual and modeled peaks, the greater the fit error %, leading to less accuracy in metabolite concentrations. Considering that GABA+ concentrations were determined in references to unsuppressed H20, FWHM, SNR, and fit error % for both GABA+ and H20 were evaluated.”

  • Moreover, in the Methods section, we also did our best to describe all the statistical parameters used in this study, although reproducibility statistics are generally used in various scientific fields. The statistical parameters are described on page 15, lines 1020-1035, as follows:

“Pearson’s correlation r measures linear correlation. It measures the strength and directions of correlation. The power of association was ranked as either very small (0 < r < 0.1), small (0.1 < r < 0.29), medium (0.3 < r < 0.49), or large (0.5 < r < 1.0) [51]. CV% is a measure of repeatability, indicating within-subject variance. It was calculated by (M/SD) x 100%, where M represented the average of a subject’s two test-retest scans [52]. Conversely, ICC assessed reliability that considers both within- and be-tween-subjects variance. In this analysis, SPSS was set to consider the single-rating, absolute agreement for two-way mixed effects ICC [53]. The quality of ICC parameters was determined according to the following convention: poor (ICC < 0.4), moderate (0.4 < ICC < 0.59), good (0.6 < ICC < 0.74), and excellent (ICC > 0.75) [52]. Moreover, the range of agreement between two test-retest scans (e.g., B2B or D2D) was visualized with Bland-Altman plots [54]. Quality parameters (GABA+ and H20 FWHM, GABA+ and H20 SNRs, GABA+ and H20 fit error %) were further examined for Bland-Altman GABA+ outliers to assess their effect on this data. The effects of differences in voxel overlaps and tissue heterogeneity on D2D reproducibility were evaluated by computing the correlation between the D2D percentage of voxel overlaps and the D2D re-producibility matrix.”

Thank you again for reviewing our manuscript.  Please do not hesitate to let us know, should you need further clarification regarding our responses.

Sincere regards,

Sonja Elsaid on behalf of all authors
